# Effects of Insect Cuticular Compounds on Appressorium Formation and Metabolic Activity in *Beauveria bassiana*

**DOI:** 10.3390/jof11120833

**Published:** 2025-11-25

**Authors:** Jiarui Chen, Huaxin Cai, Canxia Wu, Dongxu Wang, Jingyang Ni, Songqing Wu, Yinghua Tong

**Affiliations:** Forestry College, Fujian Agriculture and Forestry University, Fuzhou 350002, China

**Keywords:** insect cuticular compounds, *Beauveria bassiana*, appressorium, metabolism

## Abstract

The rate of appressoria formation following conidial germination in *Beauveria bassiana* is closely associated with its pathogenicity. This study investigated the effects of insect cuticular compounds on the formation and metabolism in *B. bassiana* through the addition of insect cuticle analogues. Results indicate that both the fatty acid compound carnitine C3:0 and the organic acid compound Thiamine Pyrophosphate (TPP) exert dose-dependent bell-shaped effects on *B. bassiana* spore germination and appressorium formation at different concentrations. Both low and high concentrations inhibit spore germination and appendage formation. At a concentration of 0.10 mg/mL, spore germination and appendage formation rates peaked at all time points, being significantly higher than the control (*p* ≤ 0.05). Compounds in the benzene and its derivatives class, such as enilconazole and disulfide bis(2-hydroxy-3-methylpropan-2-yl) (DSBA), significantly reduced spore germination and appressorium formation in *B. bassiana* (*p* ≤ 0.05), with stronger inhibition becoming more pronounced at higher concentrations. In contrast, amino acids and their metabolites (e.g., glycylmethionine) and glycerophospholipid compounds like 1,2-Dioleoyl-sn-Glycero-3-Phosphocholine (DOPC) had no significant effects on spore germination or appressorium formation at any tested concentration (*p* > 0.05). LC-MS analysis revealed that the insect cuticular fatty acyl compound carnitine C3:0 broadly modulated the secondary metabolism of *B. bassiana*. Following appressorium formation, 146 metabolites with significant changes in abundance were identified. Before appressorium formation, carnitine C3:0 promoted the activation of *B. bassiana* signaling pathways, such as Rap1, and stimulated antibiotic biosynthesis (penicillin and cephalosporin), thereby suppressing competing microorganisms and facilitating initial attachment. After appressorium formation, carnitine C3:0 activated pathways related to metabolite synthesis (e.g., arginine and nucleotides biosynthesis) and population regulation (ferroptosis), thereby enhancing appressorium function and structural stability. Thus, carnitine C3:0 enhances *B. bassiana*’s ability to establish infection sites before appressorium formation through antibiotic clearance and signal activation, and maintain infection structures after formation via metabolic reinforcement and population regulation. This study lays a theoretical foundation for further investigations into *B. bassiana* infection mechanisms, pathogenicity, and the role of its conidiophores.

## 1. Introduction

*Beauveria bassiana* belongs to the kingdom Fungi, phylum Ascomycota, subphylum Pezizomycotina, class Sordariomycetes, subclass Hypocreomycetidae, order Hypocreales, family Cordycipitaceae, and genus *Beauveria*. *B. bassiana* exhibits a broad host range, infecting more than 700 insect species from 15 orders and 149 families, as well as over 10 tick species from 6 families [1]. It has been widely used as a biological control agent against agricultural and forest pests such as pine caterpillar (*Dendrolimus*) [2], *Hyphantria cunea* [3], *Monochamus alternatus* [4], *Basilepta melanopus* [5], *Ostrinia furnacalis* [6] and other pests.

Research has shown that during successful infection of hosts by *B. bassiana*, appressorium formation is significantly and positively correlated with pathogenicity [7]. *B. bassiana* conidia adhere to the insect cuticle and germinate to form a germ tube, which develops an appressorium at its tip; from this appressorium, an infection peg subsequently emerges. Subsequently, through the synergistic action of mechanical pressure and hydrolytic enzymes, the fungus penetrates the insect cuticle and invades the hemocoel [8]. The formation of appressorium not only serves as a prerequisite for successful colonization of the host cuticle by *B. bassiana* [9], but also triggers vigorous metabolic activity within the fungus. This metabolic activity leads to the production of proteases, chitinases, lipases, and other enzymes that degrade the insect cuticle, while the infection peg simultaneously exerts mechanical pressure to facilitate cuticle penetration and ensure successful infection [9]. Luo et al. reported that during the formation of the appressorium, *B. bassiana* produces elevated levels of acylcarnitines and phospholipids, which maintain high turgor pressure within the appressorium and provide energy for host penetration [10]. Most previous studies have focused on the chemical interactions between appressoria and the insect cuticle, as well as on the physicochemical factors influencing appressorium formation in *B. bassiana* [11,12,13,14]. However, the infection process of *B. bassiana* is a complex chemical interaction that is also influenced by the chemical composition of host cuticular compounds. Previous research has shown that after *B. bassiana* infects *Opisina arenosella*, 61 differential cuticular compounds are specifically associated with the appressorium formation stage [7]. From these 61 compounds, 5 major classes with relatively high abundance were selected: fatty acyls, glycerophospholipids, amino acids and their metabolites, benzene and its derivatives, and organic acids and their derivatives. Six insect cuticular compound analogues showing significant differential abundance and potential influence on *B. bassiana* appressorium formation and metabolism were chosen for further evaluation. Their effects on appressorium formation were assessed, and untargeted LC-MS-based metabolomic analysis was performed to investigate how these analogues affect the metabolism and metabolites of *B. bassiana* after appressorium formation. This study provides a theoretical foundation for elucidating the pathogenic mechanism of *B. bassiana* and its chemical interactions with the insect cuticle.

## 2. Materials and Methods

### 2.1. Test Strains

The test strain was *B. bassiana* bcm-01, preserved and maintained in the Forest Protection Teaching and Research Section of Fujian Agriculture and Forestry University, Fuzhou, China. Following three sucessive passages through *O. arenosella*, the strain was subsequently inoculated onto Potato Dextrose Agar (PDA) medium (200 g peeled potatoes, 20 g glucose, 15–20 g agar, 1000 mL water, natural pH) and incubated at 26 °C under constant temperature.

### 2.2. Reagents

The insect cuticle analogues used in this study that may affect the spore germination and appressorium formation of *B. bassiana* include Propionyl-L-carnitine (carnitine C3:0, 99.50%), 1,2-Dioleoyl-sn-Glycero-3-Phosphocholine (DOPC, 99.50%), Glycylmethionine (99.50%), Thiamine Pyrophosphate (TPP, 99.50%), ophosphate, (99.50%), Enilconazole (99.50%), 4-Dodecylbenzenesulfonic acid (DSBA, 99.5%) All reagents from GlpBio (Montclair, CA, USA).

### 2.3. Determination of Conidial Germination Rates and Adhesive Spore Formation Rates of B. bassianan in Analogues of Different Insect Cuticular Compounds

Sterile aqueous solutions containing 1 mg/mL trehalose were prepared. Six insect cuticular compound analogues were individually added to the solutions to establish five concentration gradients for each compound. *B. bassiana* spore powder was added to each mixture to obtain a final concentration of 5 × 10^7^ spores/mL. A control was prepared under identical conditions without the addition of compound analogues. The mixtures were shaken at 150 rpm and incubated at 26 ± 1 °C under constant temperature. Samples (100 µL) were collected after 12, 24, 36, 48, and 72 h of incubation. Each sample was examined microscopically to determine germination and appressorium formation rates. Five randomly selected fields were examined per replicate, and 20 spores were counted per field. Each compound analogue was tested at a single concentration per treatment, with five biological replicates.

### 2.4. Metabolic Effects of Insect Cuticle Compound Analogues on B. bassiaan Appressorium Formation

#### 2.4.1. Sample Preparation

A sterile aqueous solution containing 1 mg/mL yeast extract was prepared, and *B. bassiana* conidial powder was added to obtain a final concentration of 5 × 10^6^ conidia/mL. An insect cuticular compound analogue that showed a pronounced effect on appressorium formation was added to the suspension, while a conidial suspension without the analogue served as the control. The mixtures were shaken at 150 rpm and incubated at 26 ± 1 °C under constant temperature. Based on the observations in Section 2.3, fungal cultures were collected both before and after appressorium formation. Each sample was filtered through a 0.22 µm membrane filter, and 1000 µL of the filtrate was transferred to an EP tube. Then 400 µL of 80% methanol (*v*/*v*) was added, vortex-mixed, and the mixture was kept on ice for 5 min. The samples were centrifuged for 20 min at 15,000× *g* and 4 °C. The resulting supernatant was diluted with LC–MS grade water to reach a methanol content of 53% (*v*/*v*) and centrifuged again under the same conditions. After three rounds of centrifugation and dilution to 53% methanol, the final supernatant was collected for LC–MS analysis [15].

#### 2.4.2. LC-MS Analysis

Samples obtained in Section 2.4.1 were analyzed using a Vanquish ultra-high-performance liquid chromatography (UPLC) system (Thermo Fisher Scientific, Waltham, MA, USA) equipped with a Waters ACQUITY UPLC BEH Amide column (2.1 mm × 100 mm, 1.7 µm). The mobile phase consisted of solvent A (aqueous 25 mM ammonium acetate with 25 mM ammonia) and solvent B (acetonitrile). The sample tray was set at 4 °C with an injection volume of 2 µL.

Mass spectrometry detection was performed using electrospray ionization (ESI) in both positive and negative ion modes. The scan range was set to *m*/*z* 100–1500. The ESI source parameters were as follows: spray voltage: 3.5 kV; sheath gas flow rate: 35 psi; auxiliary gas flow rate: 10 L/min; capillary temp: 320 °C; S-lens RF level: 60; auxiliary gas heater temperature: 350 °C; polarity: positive, negative; and MS/MS acquisition was operated in data-dependent scans.

### 2.5. Data Processing

Data were organized in Excel 2020 (Microsoft Corporation, Redmond, WA, USA) and analyzed using SPSS 25.0 (IBM, IBM Corporation, Armonk, NY, USA). Multiple comparisons among treatments with different reagent concentrations and the control were conducted using Duncan’s multiple range test. Raw data on *B. bassiana* secondary metabolites were subjected to format conversion and metabolite annotation, followed by orthogonal partial least squares discriminant analysis (OPLS-DA) [16] and Kyoto Encyclopedia of Genes and Genomes (KEGG) pathway enrichment analysis [17,18,19,20,21]. The calculation formulas are as follows:(1)$$\begin{array}{cc} \mathrm{Germination}\;\mathrm{rate}\;\left(\%\right) & =\frac{\mathrm{Number}\;\mathrm{of}\;\mathrm{germinated}\;\mathrm{conidia}}{\mathrm{Total}\;\mathrm{number}\;\mathrm{of}\;\mathrm{conidia}}\times 100 \end{array}$$(2)$$\begin{array}{cc} \mathrm{Appressorium}\;\mathrm{formation}\;\mathrm{rate}\;\left(\%\right) & =\frac{\mathrm{Number}\;\mathrm{of}\;\mathrm{conidia}\;\mathrm{forming}\;\mathrm{appressoria}}{\mathrm{Number}\;\mathrm{of}\;\mathrm{germinated}\;\mathrm{conidia}}\times 100 \end{array}$$

## 3. Result

### 3.1. Effects of Different Insect Cuticular Compound Analogues on Spore Germination and Appressorium Formation in B. bassiana

Statistical analysis of spore germination and appressorium formation rates of *B. bassiana* treated with different insect cuticular compound analogues is presented in (Appendix A). A bell-shaped response pattern was observed for the fatty acyl compound carnitine C3:0, showing maximal promotion of spore germination and appressorium formation at 0.10 mg/mL. (*p* ≤ 0.05). Both lower (0.05 mg/mL) and higher concentrations (0.50 and 1.00 mg/mL) reduced these rates, while 0.25 mg/mL showed no significant difference compared with the control (*p* > 0.05). Similarly, the organic acid compound TPP exhibited a concentration appressorium formation. At 0.10 mg/mL, the rates of both parameters reached their peak values, significantly exceeding those of the control (*p* ≤ 0.05). In contrast, benzene derived enilconazole and disulfosuccinol significantly inhibited both spore germination and appressorium formation in *B. bassiana* (*p* ≤0.05), with inhibition strength increasing with concentration. Amino acid derivatives glycylmethionine and glycerophospholipids DOPC showed no significant effects on either spore germination or appressorium formation across all concentrations tested (*p* > 0.05). These results indicate that the insect fatty acid epidermal compound carnitine C3:0 effectively promotes conidial germination and appressorium formation in *B. bassiana*.

### 3.2. Effects of the Insect Cuticle Compound Analogue Carnitine C3:0 on the Metabolism of B. bassiana Adhesive Spores

#### 3.2.1. Orthogonal Partial Least Squares Discriminant Analysis (OPLS-DA)

Metabolic profiles of *B. bassiana* were compared between the treatment group supplemented with the fatty acyl insect cuticular compound analogue carnitine C3:0 and the control group, both before and after appressorium formation. The OPLS-DA score plot (Figure 1) revealed clear separation between the treatment and control samples at both developmental stages, indicating that the addition of carnitine C3:0 markedly altered the secondary metabolite landscape of *B. bassiana*. The OPLS-DA model demonstrated excellent predictive performance (Q^2^ > 0.5), and its statistical validity was confirmed by cross-validation ANOVA (*p* < 0.05). These findings indicate that the fatty acyl compound carnitine C3:0 induces extensive reorganization of secondary metabolites in *B. bassiana*. The validated OPLS-DA model provides a reliable basis for identifying and characterizing metabolites that are differentially enriched during appressorium development.

#### 3.2.2. Volcano Analysis

The overall distribution of differential secondary metabolites between the *B. bassiana* treatment group supplemented with Carnitine C3:0 and the control group, both before and after appressorium formation, was visualized using volcano plots (Figure 2), with metabolites meeting the thresholds of a fold change > 2.0 and a *p* < 0.05 (Student’s *t*-test) considered statistically significant. Before appressorium formation, a total of 111 differential secondary metabolites were identified between the carnitine C3:0 treatment and the control group, of which 38 were significantly upregulated and 73 were significantly downregulated. Following appressorium formation, 180 differential secondary metabolites were detected, including 65 significantly upregulated and 115 significantly downregulated metabolites. These findings indicate that significant metabolic differences occurred between the treatment and control groups at both developmental stages, demonstrating that the fatty acyl compound carnitine C3:0 exerts broad regulatory effects on the secondary metabolism of *B. bassiana*.

#### 3.2.3. Differential Metabolites Following *B. bassiana* Spore Formation in the Carnitine C3:0-Treated Group

A Venn diagram was generated to compare the differential secondary metabolites of *B. bassiana* before and after the appressorium formation between the carnitine C3:0 treatment and control group (Figure 3). As shown in the figure, 111 differential metabolites were identified before appressorium formation and 180 after appressorium formation. Among these, 77 metabolites were unique to the pre-appressorium stage, 146 were unique to the post-appressorium stage, and 34 metabolites were shared between the two stages.

The 146 unique differentially expressed metabolites after appressorium formation are summarized in Figure 4. Upregulated metabolites were mainly classified as amino acids and their derivatives, benzene derivatives, glycerophospholipids, carbonyl compounds (including aldehydes, ketones, and esters), and organic acids and their derivatives. Amino acids and their derivatives represented the most enriched category (21%), including L-leucine-L-alanine, L-cysteine, cyanophosphin, threonine-isoleucine, and isoleucine-lysine. Benzene derivatives accounted for 17%, represented by amlodipine, nadolol, dimethindene, and avandrol. Unique downregulated differential metabolites primarily included amino acids and their metabolites, organic acids and their derivatives (25%), organic acids and their derivatives (13%), and benzene and its derivatives (12%). The major downregulated compounds included L-proline, L-citrulline, O-acetyl-L-serine, and N-ethylglycine, including 12 types such as iminodiacetic acid, piperidonic acid, 5-hydroxyvaleric acid, and butanedioic acid; Benzene and its derivatives account for 12%, primarily comprising 1-hydroxyamino-2-phenylethane, 4-hydroxybenzaldehyde, oryzanol, among 11 others.

#### 3.2.4. KEGG Analysis

As shown in Figure 5, in the *B. bassiana* strain treated with the fatty acyl compound carnitine C3:0, differential secondary metabolites before appressorium formation were enriched in 20 metabolic pathways compared with the control group. 18 of these pathways were uniquely enriched relative to the post-appressorium stage, mainly involving activation-related pathways such as Rap1 signaling and antibiotic biosynthesis (penicillin and cephalosporin biosynthesis), which contribute to the suppression of microorganisms and initiation of attachment. After appressorium formation, differential metabolism between the treatment and control groups were enriched in 20 metabolic pathways. Compared with the pre-appressorium stage, 18 unique metabolic pathways were identified, mainly associated with enhanced metabolic biosynthesis (arginine and nucleotides) and population regulation (ferroptosis), promoting the functional optimization and structural stability of appressoria. Two pathways—the two-component system and quorum sensing—were commonly enriched in both stages, indicating coordinated regulation between metabolism and environmental responses. These demonstrate that the addition of fatty acyl carnitine C3:0 optimizes the growth efficiency of *B. bassiana* compared with the control group. The differential enrichment of metabolic pathways indicates that carnitine C3:0 enhances the fungus’s ability to establish infection sites before appressorium formation (through antibiotic clearance and signal activation) and to maintain infection structures after formation (through metabolic reinforcement and quorum regulation).

## 4. Discussion

This study found that insect cuticular compound analogues significantly affected the germination and appressorium formation rates of *B. bassiana*. Among them, the insect cuticular fatty acyl compound carnitine C3:0, at a concentration of 0.10 mg/mL, significantly promoted spore germination and appressorium formation (*p* ≤ 0.05). As specialized infection structures, appressoria penetrate the host cuticle via infection pegs or invasive hyphae and by secreting extracellular enzymes such as proteases, chitinases, and lipases [22,23]. The appressorium formation rate of *B. bassiana* is significantly and positively correlated with its pathogenicity [7]. These results provide a theoretical basis for enhancing the virulence of fungal insecticides. As a typical entomopathogenic fungus, the formation of appressoria in *B. bassiana* marks the maturation of its infection structures. The differential metabolite profiles at this stage can reveal mechanisms underlying its virulence regulation [24]. In this study, LC-MS was employed for metabolomics analysis. After the addition of carnitine C3:0, 111 differentially expressed metabolites were identified in *B. bassiana* before appressorium formation compared with the control group. These metabolites were mainly enriched in pathways associated with signal activation (Rap1 signaling) and antibiotic biosynthesis (penicillin and cephalosporin biosynthesis), indicating the suppression of competing microorganisms and initiate adhesion. This aligns with broader fungal signal transduction frameworks, where Rap1 signaling acts as a key integrator of extracellular cues, such as nutrient availability and surface hydrophobicity, to trigger morphological changes essential for host penetration [25,26]. Such mechanisms are critical for environmental adaptation in entomopathogenic fungi, allowing them to sense and respond to variable host defenses and abiotic stresses like desiccation or UV exposure, thereby optimizing virulence in diverse ecological niches. After appressorium formation, 180 differential metabolites were compared between the carnitine C3:0-treated and control groups. These metabolites were predominantly enriched in pathways related to enhanced metabolite synthesis (arginine and nucleotide biosynthesis) and population regulation (ferroptosis), contributing primarily to the maintenance of infection structures through metabolic reinforcement and population regulation. This observation is consistent with previous findings that carnitine participates in regulating intracellular fatty acid metabolism, including propionic acid produced by its own metabolism, thereby influencing energy supply and lipid-associated signaling pathways, ultimately promoting fungal growth and development [27,28]. Furthermore, these metabolic shifts reflect conserved strategies in fungal environmental adaptation, where signal transduction pathways modulate resource allocation to sustain infection under nutrient-limited or oxidative stress conditions, as seen in other pathogens like *Metarhizium anisopliae* [29]. By integrating carnitine-mediated regulation into these broader networks, our results highlight how *B. bassiana* enhances its adaptive plasticity, ensuring successful pathogenesis in fluctuating environments.

A total of 146 unique differential metabolites were identified after appressorium formation in *B. bassiana*, including organic acids and their derivatives, aromatic compounds, and amino acid-related metabolites. Organic acids and their derivatives play essential roles in the growth of *B. bassiana* by regulating the pH of the culture medium, chelating metal ions, and supplying essential trace elements, thereby promoting hyphal growth and spore formation [28,30,31,32]. Aromatic compounds may act as signaling molecules that regulate the growth and metabolic process of *B. bassiana* by activating or repressing specific gene expression and modulating metabolic pathways. In addition, these compounds may exhibit antimicrobial activity, enabling *B. bassiana* to compete more effectively for resources in natural environments [33,34,35]. Amino acids and their derivatives are essential nutrients for *B. bassiana*, serving as nitrogen sources that participate in protein synthesis and promote hyphal growth and spore formation [36,37,38]. Certain amino acids and their derivatives may also act as signaling molecules that activate specific signaling pathways, thereby promoting the differentiation and maturation of appressoria [39,40,41]. These findings indicate that carnitine C3:0 markedly influences the secondary metabolism of *B. bassiana* during the appressorium formation stage. The functions of the following four differentially expressed metabolites warrant further investigation: Among these metabolites, piperidine, a six-membered nitrogen-containing heterocyclic scaffold, may enhance the virulence of *B. bassiana* through dual mechanisms. (1) It may serve as a precursor for the biosynthesis of piperidine alkaloids (e.g., anabasine) that possess broad-spectrum antibacterial activity, thereby disrupting the membrane integrity of competing microorganisms, inhibiting their growth, and reducing microbial competition. (2) It may also give rise to insecticidal toxins such as 2-piperidone, which interfere with the host’s normal physiological functions and thereby facilitate fungal infection [42]. Oxalic acid derivatives, such as oxalic acid and calcium oxalate, may enhance *B. bassiana* through the following mechanisms: (1) By strongly acidifying the environment (pH 2–3), these compounds dissolve the chitin-protein complex in the insect cuticle, thereby softening the body wall structure and creating physical pathways for hyphal invasion [43]. (2) By competitively inhibiting key enzymes in the TCA cycle, such as succinate dehydrogenase, they block ATP production, leading to host muscle paralysis and organ failure [44]. In additionally, D-kynurenine, a metabolite of tryptophan, may enhance fungal virulence during infection of host insect through two pathways: (1) Kynurenine exhibits neurotoxic effects on adult insects, causing severe motor dysfunction and prolonging the window of opportunity for *B. bassiana* to penetrate the cuticle [45]. (2) D-kynurenine inhibits the expression of βGRP (β-glucan recognition protein) in the Toll signaling pathway, impairing the host immune recognition of fungal cell wall components, reducing the efficiency of antimicrobial peptide synthesis, and weakening the insect’s immune defence against *B. bassiana* [46]. Polyketide compounds may enhance the virulence of *B. bassiana* through the following mechanisms: (1) Suppressing cellular and humoral immune responses in insects, thereby facilitating pathogen evasion of host defences [47]. (2) Upregulating the biosynthesis of other toxins such as beauvericin and its derivatives, to further amplify fungal virulence [48]. (3) Maintaining normal conidial morphology and cell wall integrity, thereby improving resistance to immune effector molecules such as reactive oxygen species (ROS) in the insect hemolymph [49]. The insect cuticular fatty acyl compound carnitine C3:0 was found to induce *B. bassiana*, which promotes infection. The metabolic alteration observed in this study suggests a triple mechanism: (1) carnitine C3:0 facilitates fungal invasion by stimulating the biosynthesis of antimicrobial metabolites, which suppress the growth of host symbiotic bacteria and establish a microenvironment conducive to infection; (2) carnitine C3:0 promotes the production of toxins or acidic metabolites that disrupt host physiological functions, thereby enhancing fungal penetration and infection efficiency; (3) carnitine C3:0 enhances the synthesis of immunosuppressive metabolites, interfering with the normal operation of the host immune system and weakening the host’s defense against *B. bassiana*.

## 5. Conclusions

The results of this study indicate that (1) at specific concentrations, the insect cuticular fatty acyl compound carnitine C3:0 and the organic acid compound thiamine pyrophosphate (TPP) promote spore germination and appressorium formation in *B. bassiana*, thereby facilitating host infection; and (2) the effect of the cuticular fatty acyl compound carnitine C3:0 on *B. bassiana* involves not only energy supply but also complex chemical signaling and metabolic interference. The underlying mechanism by which carnitine C3:0 regulates spore germination, appressorium formation, and virulence enhancement warrants further investigation. This study provides new perspectives for exploring fungus–insect cuticle interactions and establishes a theoretical basis for elucidating the pathogenic mechanisms of *B. bassiana* and optimizing its application in biological control.

## Figures and Tables

**Figure 1 jof-11-00833-f001:**
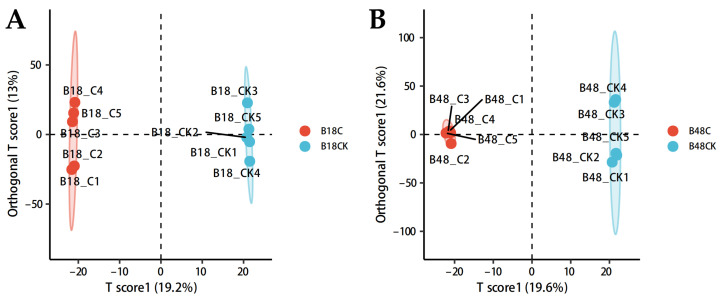
OPLS-DA score plots showing metabolic distinctions between groups. (**A**) Comparison between B18_C and B18_CK groups, (**B**) Comparison between B48_C and B48_CK groups. Note: The scatter plots reveal clear inter-group separation along the predictive component (x-axis) and intra-group dispersion along the orthogonal component (y-axis). Ellipses represent the 95% confidence interval (Hotelling’s T^2^). Model validity is supported by cross-validation parameters, with Q^2^ > 0.5 indicating robust predictive capability.

**Figure 2 jof-11-00833-f002:**
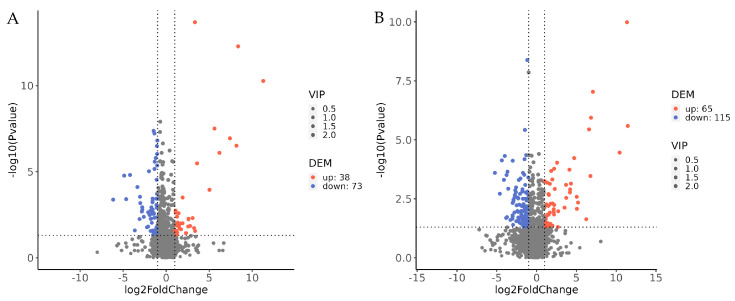
Volcano plots illustrating differential metabolite accumulation between experimental groups. (**A**) Differential expression profile for the B18 comparison, (**B**) Differential expression profile for the B48 comparison. Note: Each dot represents a distinct metabolite. Significantly upregulated (red) and downregulated (blue) metabolites were identified based on variable importance in projection (VIP > 1) and statistical significance (*p* < 0.05, −log10 transformed). Gray dots represent metabolites that were not identified as differential metabolites. The vertical dotted line marks the boundary of no difference in expression, while the horizontal dotted line indicates the statistical significance threshold (−log10 *p*-value). Point size is proportional to the VIP score. In order to show the distribution of the differences more clearly, the Y-axis range of (**B**) is adjusted to (0, 10).

**Figure 3 jof-11-00833-f003:**
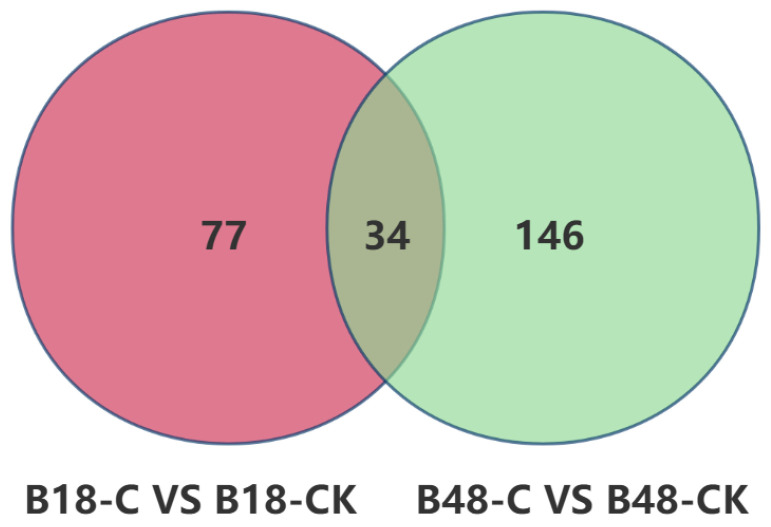
Venn diagrams illustrating the overlapping and distinct differential metabolites of *B. bassiana* under carnitine C3:0 treatment before and after appressorial formation, compared with time-matched controls.

**Figure 4 jof-11-00833-f004:**
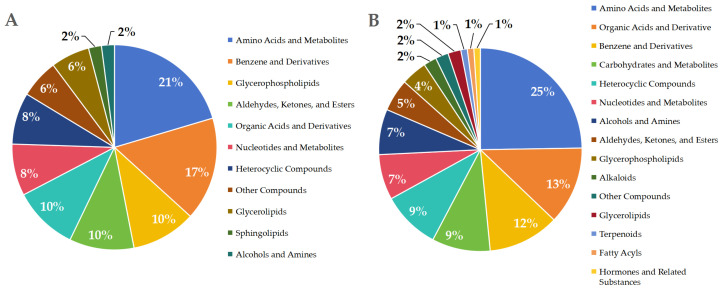
Categories of differential metabolites uniquely enriched in the carnitine C3:0-treated group after appressorial formation. (**A**) upregulated differential metabolites, (**B**) downregulated differential metabolites.

**Figure 5 jof-11-00833-f005:**
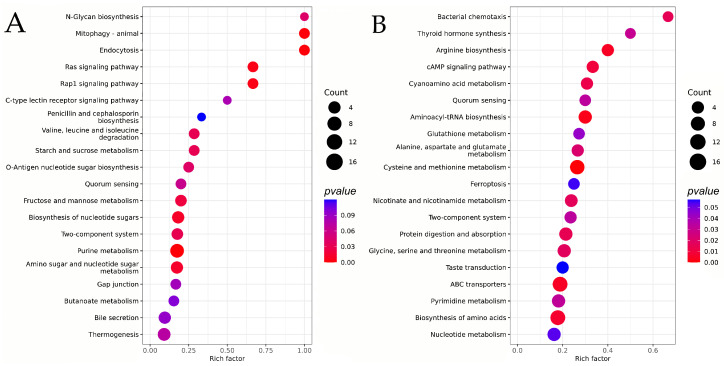
Enrichment analysis of differential secondary metabolites in the carnitine C3:0-treated group before and after appressorial formation compared with time-matched controls. (**A**) Before appressorium formation, (**B**) following appressorium formation.

## Data Availability

The original contributions persented in this study are included in the article and Appendix A. Further inquiries can be directed to the corresponding author.

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
