# Peer review of "Effects of Insect Cuticular Compounds on Appressorium Formation and Metabolic Activity in *Beauveria bassiana"

_jof, 2025, doi:10.3390/jof11120833_

Round 1
Reviewer 1 Report
The manuscript titled “Effects of Insect Cuticular Compounds on Appressorium Formation and Metabolic Activity in Beauveria bassiana” presents a detailed investigation into how insect cuticular compound analogues influence spore germination, appressorium formation, and metabolic profiles of Beauveria bassiana. The study combines morphological assays with LC–MS-based metabolomics to elucidate chemical and metabolic mechanisms underlying fungal pathogenicity. Overall, the research is original, scientifically sound, and potentially valuable for understanding fungus–insect interactions relevant to biocontrol.
Major Weaknesses and Recommendations
-
Language and Style
-
The manuscript would benefit from linguistic editing to correct typographical and grammatical inconsistencies
-
Minor redundancies appear in the Abstract and Discussion (e.g., duplicated phrases like “at any tested concentration at any concentration”).
-
-
Statistical Reporting
-
While Duncan’s multiple range test is used, the authors should include F-values or ANOVA summaries in the main text or supplementary section to improve transparency.
-
For metabolomic results, volcano plot thresholds (fold change and p-value) should be explicitly defined in the methods.
-
-
Discussion Depth
-
The discussion is strong but could further contextualize findings within the broader framework of fungal signal transduction and environmental adaptation.
-
Some speculative connections (e.g., proposed neurotoxic or immunosuppressive roles of certain metabolites) require caution or supporting citations
-
Minor Comments
-
Line 78-80: Add hypothesis.
- Line 99: Correct 5 × 107 spores/mL (and bellow the text)
-
Line 232–245: Merge the discussion on enzyme secretion and host penetration with prior mechanistic context for smoother flow.
-
Ensure all abbreviations (e.g., TPP, DSBA, DOPC) are defined at first use.
Reviewer 2 Report
Effects of Insect Cuticular Compounds on Appressorium Formation and Metabolic Activity in Beauveria bassiana
The document is well-written and contains valuable information to help understand the infection mechanisms of B. bassiana in insect pests. The title includes "the appressorium formation," so it is recommended that it be illustrated with images as evidence of this process. The explanation of cuticular compounds is well-defined, and clear and precise evidence is provided.
Line 7. "Chiana" Is it correct?
INTRODUCTION
The introduction should describe the appressorium formation process and explain whether C3:0 carnitine, in addition to modifying the production of cuticular compounds, forms more rapidly, thus shortening the duration of the infectious process. Explain what it means that C3:0 carnitine enhances the ability to establish infection. Further examples of insects infected by B. bassiana should also be mentioned, and it should be explained whether the infection shares a common mechanism in all cases and whether C3:0 carnitine can have the same effect.
Line 67. Space between ofB.
Line 73. Write "6" in word "Six"
MATERIALS AND METHODS
Line 84. After the name of the Institution, add the city and country.
Line 85. When a number has only one digit, it is recommended to write it out in words.
Lines 96, 97, 103. When a number has only one digit, it is recommended to write it out in words.
RESULT
The legends for the figures need improvement in the results section.
Lines 148-149. Delete the sentence, it goes without saying.
Figure 1. It is recommended that both figures A and B, have the same scale; this allows the dimension of the differences to be appreciated.
Figure 2. It is recommended that both figures, A and B, have the same scale; This allows the dimension of the differences to be appreciated.
Figures 3, 4 and 5. Edit all legends and titles in black because in the figures they appear to be in shades of gray. The text inside the figures is ok.
